# Potentially Modifiable Factors Associated with Adherence to Adjuvant Endocrine Therapy among Breast Cancer Survivors: A Systematic Review

**DOI:** 10.3390/cancers13010107

**Published:** 2020-12-31

**Authors:** Kirsti I. Toivonen, Tamara M. Williamson, Linda E. Carlson, Lauren M. Walker, Tavis S. Campbell

**Affiliations:** 1Department of Psychology, University of Calgary, Calgary, AB T2N 1N4, Canada; kirsti.toivonen@ucalgary.ca (K.I.T.); tamara.williamson@ucalgary.ca (T.M.W.); 2Department of Oncology, University of Calgary, Calgary, AB T2N 1N4, Canada; l.carlson@ucalgary.ca (L.E.C.); lauren.walker@albertahealthservices.ca (L.M.W.); 3Department of Psychosocial Resources, Tom Baker Cancer Centre, Holy Cross Site, Calgary, AB T2S 3C1, Canada

**Keywords:** review, adherence, breast cancer, adjuvant endocrine therapy

## Abstract

**Simple Summary:**

Endocrine therapy taken after primary breast cancer treatment helps prevent breast cancer recurrence. However, many women are unable to adhere to endocrine therapy. This review examines potentially modifiable factors that are associated with endocrine therapy adherence, which might help future efforts to improve endocrine therapy use. Six categories of factors were identified: side effects, attitudes toward endocrine therapy, psychological factors, healthcare provider-related factors, sociocultural factors, and general or quality of life factors. Overall, self-efficacy (i.e., one’s belief in their ability to do something) and positive decisional balance (i.e., one’s belief that the benefits of endocrine therapy outweigh the risks) were the most consistently associated with adherence. They might represent factors worth investigating in future studies seeking to support the adherence of breast cancer survivors.

**Abstract:**

Adjuvant endocrine therapy (AET) reduces risk of breast cancer recurrence. However, suboptimal adherence and persistence to AET remain important clinical issues. Understanding factors associated with adherence may help inform efforts to improve use of AET as prescribed. The present systematic review examined potentially modifiable factors associated with adherence to AET in accordance with PRISMA guidelines (PROSPERO registration ID: CRD42019124200). All studies were included, whether factors were significantly associated with adherence or results were null. This review also accounted for the frequency with which a potentially modifiable factor was examined and whether univariate or multivariate models were used. This review also examined whether methodological or sample characteristics were associated with the likelihood of a factor being associated with AET adherence. A total of 68 articles were included. Potentially modifiable factors were grouped into six categories: side effects, attitudes toward AET, psychological factors, healthcare provider-related factors, sociocultural factors, and general/quality of life factors. Side effects were less likely to be associated with adherence in studies with retrospective or cross-sectional than prospective designs. Self-efficacy (psychological factor) and positive decisional balance (attitude toward AET) were the only potentially modifiable factors examined ≥10 times and associated with adherence or persistence ≥75% of the time in both univariate and multivariate models. Self-efficacy and decisional balance (i.e., weight of pros vs. cons) were the potentially modifiable factors most consistently associated with adherence, and hence may be worth focusing on as targets for interventions to improve AET adherence among breast cancer survivors.

## 1. Introduction

Adjuvant endocrine therapy (AET), including the selective estrogen receptor modulator tamoxifen and aromatase inhibitors (AIs), is well established to reduce risk of hormone receptor-positive breast cancer recurrence [1,2,3,4]. As part of standard care, AET is recommended for 5–10 years following primary treatment (surgery, chemotherapy, radiation therapy) [5,6]. Important clinical issues relating to AET use include treatment adherence (i.e., the extent to which a person conforms to a medication’s prescribed dose, frequency, and duration [7]) and persistence (i.e., the duration of medication use [7]). Specific estimates of AET adherence range from 75% to over 90% but vary by measure used and timeframe examined. For example, a systematic review reports higher estimates of adherence based on Medication Event Monitoring System data (93%) than on self-report (82%) and prescription refill data (75%) [8]. The same review reports average estimates of adherence ranging from 79% in the first year of use to 56% in the fourth and fifth years [8]. Similarly, discontinuation rates are reported to rise from 21% in the first year of treatment to 48% in the fifth year [8]. Notably, AET non-adherence (defined by a medication possession ratio [MPR] < 80%) and discontinuation are associated with a 49% and 26% increase in all-cause mortality, respectively [9].

In recognition of the problem of suboptimal AET adherence, several behavioral trials have been developed to target their use in breast cancer survivors, with little success. Two systematic reviews collectively describe seven interventions (all included patient education, three also included reminders, one included problem solving and self-management strategies) and report that none improved adherence to AET [10,11]. Limitations of these interventions often include high baseline adherence rates (>80%) among both intervention and control groups, short follow-up periods (e.g., one year) relative to recommended duration of AET, and insufficient power to detect group differences [10,11]. Thus, it is difficult to determine whether null results are due to ineffectiveness of the interventions, limitations in study design, or both.

Several reviews identify that being unmarried, having more comorbidities, identifying as non-White, having later-stage cancer, extremes of age, and higher cost for AET are associated with poorer AET adherence [12,13,14,15,16], though these factors are non-modifiable. Multiple systematic and integrative literature reviews examine patient-reported and psychosocial factors associated with AET adherence or persistence in breast cancer survivors—they generally measure AET adherence and persistence via self-report or prescription records and define adherence as MPR ≥ 80% [8,12,13,16,17]. In reviews that only summarize significant relationships, factors associated with AET adherence include side effects, self-efficacy, belief in necessity of medications, social support, healthcare provider (HCP) relationship, forgetfulness, and knowledge of cancer [12,16]. Reviews that also include null findings typically endorse social support, positive decisional balance, beliefs about medications, and self-efficacy as associated with adherence but also indicate that patient–provider relationship or communication, depressive symptoms, and side effects are not always associated with adherence [8,17]. A 2015 meta-analysis indicates that side effect presence is associated with over five times the odds of discontinuing AET and nearly two times greater odds of non-adherence, however, this meta-analysis includes only two studies [18].

There are notable limitations to existing systematic reviews. First, reviews that only report factors that are associated with adherence [12,15,16] and omit studies with null findings may overestimate the importance of these variables in understanding adherence. Second, there is substantial heterogeneity in measurement and definitions of adherence, study design, and timeframes examined. To better understand the phenomenon, these measurement variables could be investigated with respect to their influence on whether factors are associated with adherence (e.g., if side effects are more likely to be associated with adherence in cross-sectional or prospective studies). Finally, extant reviews typically describe whether or not factors are associated with adherence but provide little information about the relative importance of such factors in understanding adherence (i.e., which are most consistently related to adherence).

The present study focuses on identifying which potentially modifiable factors are most consistently associated with adherence to AETs, relative to how often they are examined. This addresses gaps in the breast cancer literature by assessing each factor and synthesizing information about the proportion of studies that report significant associations with adherence (vs. null findings), allowing for relative comparison between factors. Further, the present study examines whether methodological characteristics of studies may help explain inconsistency in factors that are reported to have positive associations with adherence. This information may help narrow the focus to potentially modifiable factors that are most promising and which warrant the time, effort, and cost to investigate within behavioral intervention research.

## 2. Methods

This review was performed in accordance with PRISMA guidelines [19] and the protocol was registered with PROSPERO (ID: CRD42019124200). Deviations from the original protocol are detailed where relevant.

### 2.1. Scope of Review

There were three inclusion criteria with respect to population: (1) being a female breast cancer survivor, though studies were not excluded if they contained a small (<5%) number of men. There were no exclusions based on cancer stage. (2) Using adjuvant endocrine treatment with either third-generation aromatase inhibitors (anastrozole, letrozole, exemestane) or tamoxifen. Studies of women considered high risk for cancer and receiving prophylactic endocrine therapy were excluded. (3) Being in a clinical practice setting. Studies were excluded if the population was part of a clinical trial examining AET efficacy to avoid potential bias favoring adherence.

Outcomes of interest included associations between potentially modifiable factors and adherence, persistence, or discontinuation. The definition of a “potentially modifiable factor” is ambiguous, for example, some factors may be technically modifiable (e.g., geographical location, cost of treatment) but not realistically modifiable within a behavioral intervention. The scope of “potentially modifiable factors” thus required refinement following the publication of the protocol through two additional inclusion criteria: (1) being feasibly modifiable at the individual level. Systemic factors related to socioeconomic status (e.g., insurance coverage) and healthcare system factors (e.g., treatment setting) were therefore excluded. (2) Only patient-reported factors were included to ensure that these factors accurately reflected patient experience. For example, studies where depression was inferred rather than self-reported (e.g., based on antidepressants in prescription records) were excluded, as antidepressants are often prescribed for other uses (e.g., anxiety disorders, insomnia, pain [20], and vasomotor symptoms [21]). With respect to adherence, although adherence, persistence, and discontinuation are separate constructs, they are herein referred to as ‘adherence’ but the specific measure used in each study is described. Studies were excluded if: (1) adherence was not a primary outcome or, (2) the only reported outcome was AET initiation/non-initiation or medication switching (as switching could be medically indicated). Any measure of adherence (e.g., self-report and prescription records) was deemed acceptable for inclusion. All studies that examined an association were included, regardless of whether significant associations or null results were reported.

Cross-sectional, prospective, and retrospective study designs were eligible for inclusion. Exclusion criteria were: (1) interventions targeting adherence, (2) case studies, (3) non-original data, (4) non-English language, and (5) not a peer-reviewed journal article. While the original protocol included qualitative studies, they were ultimately omitted during the full-text screening phase. This was to avoid adding heterogeneity to results that would have required a distinct approach to quality analysis and precluded synthesis with quantitative results.

### 2.2. Search Strategy

The following databases were included in the search: CINAHL, Ebsco, EMBASE, Medline, PsychINFO, PubMed, and Web of Science. The search encompassed three key themes: breast cancer, AETs, and adherence. Subject headings and keywords were combined through Boolean operators (OR within and AND between each theme). See Appendix A for a full example of the search strategy using in Medline. The literature search was conducted on 28 June 2019 and was limited to studies published in 1998 onwards (to coincide with publication of the Early Breast Cancer Trailists’ Collaborative Groups’ findings that tamoxifen reduces risk of breast cancer recurrence [22]). This search strategy yielded 23,392 records in total and 13,609 once duplicates were removed. See Figure 1 for the PRISMA flowchart detailing record identification, selection, and inclusion.

### 2.3. Study Selection

The first phase of screening involved two authors independently screening titles and abstracts of the 13,609 records. This phase of screening was purposely liberal, excluding only those records that clearly did not meet inclusion criteria (e.g., animal studies, non-breast cancer populations). Any record that was retained during the title/abstract screen by either author was screened again in the full-text screening phase (which included 634 records). Two authors independently screened full texts, which resulted in moderate interrater agreement (Cohen’s kappa = 0.76 [23]). Each discrepancy was resolved by the authors who performed the full-text screen through discussion, for a final number of 68 studies to be included.

### 2.4. Data Extraction

One author extracted all data and double checked all values using forms with pre-identified variables: first author, year, country, sample size, average age, unique sample characteristics, recruitment setting, study design (and length of follow-up period, if applicable), length of time on AETs, the outcome being examined, how the outcome was measured, and the proportion of the sample that adhered. Through the extraction process, variables were added to capture how authors of studies defined adherence, how many associations were examined (with potentially modifiable factors only), and whether any of the factors they were investigating were based on a theory or model of health behavior change. Modal menopausal status, cancer stage, and primary treatment were initially extracted but ultimately not included as they were often not reported. Data about potentially modifiable factors included the factor, the nature of the relationship with adherence (positive, negative, null), and the type of measure used (e.g., published questionnaire, author-created questionnaire).

To evaluate risk of bias, one study-level variable (whether sampling was consecutive, random, or population-based vs. convenience sampling) and five outcome-level variables (whether adherence was clearly defined, studies had complete outcome reporting, inclusion/exclusion criteria were clearly defined, authors statistically adjusted for at least one potential confound, and any control for multiple comparisons such as statistical corrections or use of multivariate models) were extracted.

### 2.5. Analysis Plan

Substantial heterogeneity of data precluded any meta-synthesis of results based on strength of association. Thus, a narrative synthesis approach was taken with the aim of identifying which factors were most and least frequently associated with adherence. Several studies reported results of univariate analyses, multivariate analyses (i.e., adjusting for confounds or multiple predictor variables), or both. The proportion of results for all univariate analyses and multivariate analyses were recorded, where applicable. Occasionally studies reported multivariate results only—in these instances, we assumed that significant findings would also be significant in a univariate analysis (and coded them as such) and coded non-significant multivariate findings as non-significant univariate findings. When studies examined an association between a potentially modifiable factor and multiple measures of adherence separately (e.g., intentional and unintentional non-adherence), each analysis was considered separately. The total number of potential associations examined therefore exceeds the number of studies included. Within each study, the potential association between each potentially modifiable factor and each measure of adherence was coded “yes” if the variables were related in the expected direction, and “no” if there were null findings or if authors reported a significant association in an unexpected direction. In cases where associations were examined across multiple time points or different AETs, they were coded “yes” if associated ≥50% of the time.

An exploratory aim was added since the registration of the original protocol (as meta-analysis was not possible). This aim examined whether a factor was more or less likely to be associated with adherence based on three methodological characteristics: study design (prospective vs. cross-sectional/retrospective), outcome assessed (adherence vs. discontinuation), and how outcome was measured (subjective reporting vs. objectively measured). This review also examined whether a factor was more or less likely to be associated with adherence based on three sample characteristics: average age, adherence level, and time on AET. Two-tailed Pearson Chi square tests were used for categorical methodological variables and logistic regression was used for continuous variables (significance threshold of *p* < 0.05 for both). These exploratory analyses were only undertaken with factors that were investigated at least 20 times, and only with univariate results.

## 3. Results

### 3.1. Study Characteristics

See Table 1 for summary information about the characteristics of the 68 included studies, and Appendix A for complete study characteristic information. Overall, most samples included prospective designs (*n* = 31, 45.6%), assessed use of both AIs and tamoxifen (*n* = 47, 69.1%), assessed adherence (*n* = 55, 67.9%) rather than discontinuation, measured adherence through self-report (*n* = 57, 70.4%), and recruited samples from a hospital or clinic (*n* = 44, 64.7%). Sample size ranged from 31 to 13,539 and mean age ranged from 36.9 to 72.8 years. In 28 studies, women initiated AET at the time of the study; excluding these studies, the average duration of AET use ranged from 4.5 to 36 months. The number of associations between potentially modifiable factors and adherence examined in a study ranged from 1 to 91 (only tallied for studies with complete outcome reporting), and studies typically also evaluated several additional factors that were not potentially modifiable thus beyond the scope of this study. Over half of the studies (*n* = 38) were conducted in the USA.

There was substantial variability in the rates of adherence reported, ranging from 25.7% (in a study which defined ‘adherent’ as a perfect score on a self-report questionnaire [24]) to 98% (in a study which defined ‘adherent’ as MPR ≥ 80% [25]), overall averaging 74.8%. Four of the included studies reported different measures of adherence for the same sample and noted discrepancies based on the measure used. Font et al. [26] noted self-report (defined as having no or few problems adhering, 92%) and physician-reported adherence (based on chart review, 94.7%) to be higher than adherence based on pharmacy records (proportion of days covered [PDC] ≥ 80%, 74.7%). Similarly, Ziller et al. [27] reported 79.2% based on prescription records (MPR ≥ 80%) vs. 100% based on self-report. Hadji et al. [28] reported that physician-reported adherence was higher (>95%) than self-reported adherence (<70%) when both were reporting whether ≥80% of pills were taken. Kuba et al. [29] reported that examination of pill packets yielded an estimate of 85% adherent while pharmacy records suggested 98% were adherent (both defining adherence as ≥80% of pills taken). Among studies that estimated intentional and unintentional non-adherence separately [30,31,32], unintentional non-adherence (e.g., forgetting) was more common than intentional (e.g., deciding to miss a pill). When stratified by reporting source, the estimated proportion of those adherent was lower for self-report (71.3%) relative to objectively measured (81.9%) or physician reported adherence (81.3%). The proportion who discontinued averaged 23.46%, ranging from 6% to 51.5%.

### 3.2. Study Quality Assessment

See Table 2 for summary information about study quality and Appendix A for complete information. Ninety-seven percent of studies clearly described eligibility criteria and 90% clearly defined how they measured adherence or discontinuation. Most studies (70.6%) clearly had complete outcome reporting. Often it was unclear whether a study met this criterion because the description of measures was vague; thus, it is possible that a higher proportion of studies did have complete outcome reporting. Similarly, while 44% of studies clearly recruited patients using consecutive, random, or representative sampling, recruitment details was often unclear due to insufficient reporting—thus a higher proportion of studies may have also met this criterion. Finally, 69.1% and 69.7% of studies statistically adjusted for potential confounds or controlled for multiple comparisons, respectively. Overall, the studies included in this review explicitly met an average of four quality indicators.

### 3.3. Potentially Modifiable Factors Examined

Appendix A reports complete outcome information. Following data extraction, potentially modifiable factors were grouped into six categories: side effects, attitudes toward AETs, psychological factors, healthcare provider-related factors, sociocultural factors, and general/quality of life factors. See Figure 2 for a visual representation of the overall results across the six categories. The figure resembles a target with the factors associated with adherence more often placed closer to the centre. This figure also accounts for how often factors were assessed (≥10 or ≥5 times) and the proportion of significant associations among univariate and multivariate models.

The figure demonstrates that self-efficacy (a psychological factor) and positive decisional balance (an attitude toward AET) are the only two factors that have been examined ≥10 times and are associated with adherence at least 75% of the time in both univariate and multivariate models. Negative emotions (attitude toward AET) about AET, social support (sociocultural factor), and quality of relationship with HCP (HCP-related factor) were also important, being associated with adherence ≥75% of the time in univariate models. However, these factors were either examined in less than 10 studies, not associated ≥75% of the time in multivariate models, or both. A description of the potentially modifiable factors within the six categories is outlined subsequently.

#### 3.3.1. Side Effects

A total of 44 studies investigated whether global or specific measures of side effects were associated with adherence [24,25,26,27,28,29,30,31,32,33,34,35,36,37,38,39,40,41,42,43,44,45,46,47,48,49,50,51,52,53,54,55,56,57,58,59,60,61,62,63,64,65,66,67,68]. See Table 3 for an overview of results. An additional composite measure, any measure of side effects, was created across all studies (when one study examined multiple side effects, an association would be classified as “present” if ≥50% of the side effects were associated). In the 44 studies, potential associations between any measure of side effects and adherence were examined 54 times and significant associations in the expected direction were reported approximately half the time in univariate models (this was similar in multivariate models). Side effects were also common: up to 94% endorsed side effect presence, up to 88% endorsed arthralgia (i.e., joint pain), and up to 96% endorsed vasomotor/menopausal (i.e., hot flashes, night sweats) symptoms. Side effect presence, side effect severity, number of symptoms, arthralgia, and cognitive changes represented the factors most consistently associated with adherence (≥50% of the time) in univariate models. Vasomotor/menopausal symptoms were frequently examined (16 times) but only seldom associated with adherence (19% of the time).

Studies with retrospective or cross-sectional designs were less likely to report an association between any measure of side effects and adherence than studies with prospective designs (X^2^(1) = 4.83, *p* = 0.045). The likelihood of a significant association did not differ by whether the outcome was adherence (*n* = 39) or discontinuation (*n* = 14), nor whether the outcome was subjectively reported (*n* = 45) or objectively measured (*n* = 8). It was not associated with average sample age (OR = 1.09, *p* = 0.15), adherence (OR = 1.00, *p* = 0.96), or time on AET (OR = 0.96, *p* = 0.17). Only four studies distinguished intentional from unintentional non-adherence; all reported intentional non-adherence being associated with side effects while just one reported unintentional non-adherence being associated with side effects. Just over half (55%) of studies used a validated measure of side effects.

In sum, any measure of side effects was associated with adherence approximately half of the time, and significant associations were less likely to occur in studies with retrospective or cross-sectional designs. Of specific side effects, only arthralgia and cognitive changes were associated with adherence about half the time.

#### 3.3.2. Attitudes toward AET

A total of 29 studies examined whether attitudes toward AET were associated with adherence [24,29,30,31,32,33,34,37,38,44,45,46,50,54,55,58,60,61,62,64,69,70,71,72,73,74,75,76,77] (see Table 4). Having a positive decisional balance was the only factor examined 10 times and associated ≥75% of the time in univariate and multivariate models. Belief in efficacy/necessity of AET was examined 20 times; it was associated 60% of the time in univariate models but more often (88.9%) in multivariate models. Intention to take AET, attributing side effects to AET, and expected side effect severity were most often associated with adherence (≥75%) but only examined by a few studies. The belief that AETs are harmful/overused was never associated with adherence.

The likelihood of belief in efficacy/necessity of AET being associated with adherence did not differ by whether studies had a prospective (*n* = 2) vs. cross-sectional or retrospective designs (*n* = 18), measured adherence (*n* = 17) vs. discontinuation (*n* = 1), or assessed adherence with subjective (*n* = 18) or objective measures (*n* = 1). Nor did it differ with average sample age (OR = 1.06, *p* = 0.54), adherence (OR = 0.99, *p* = 0.74), or time on AET (OR = 1.09, *p* = 0.38). Fewer than half of studies (41%) explicitly reported using a validated measure of attitudes, most (59%) used author-created or adapted measures.

In sum, a positive decisional balance was most consistently associated with adherence. Belief in efficacy/necessity of AET was associated approximately half the time and was no more likely to be associated with adherence based on any methodological characteristics.

#### 3.3.3. Psychological Factors

In total, 30 studies examined whether 17 different psychological factors were associated with adherence (see Table 5) [24,29,31,32,34,38,39,41,42,52,55,60,61,62,63,64,69,70,72,73,74,76,77,78,79,80,81,82,83,84]. Self-efficacy was the most consistent predictor of adherence, being examined 10 times and associated with adherence ≥80% of the time in univariate and multivariate models. Nearly all studies examining self-efficacy were cross-sectional (*n* = 9) and all used a subjectively reported measure of adherence. Depressive symptoms were examined 14 times and were associated with adherence over half of the time in univariate models and multivariate models. Fear of cancer recurrence and personal control over outcomes were seldom associated with adherence (25% of the time or less). Several factors (e.g., optimism, perceived ageing, and perceived cognitive function) were associated with adherence but were only examined in one study. No psychological factors were examined often enough to examine whether likelihood of being associated with adherence varied with methodological characteristics. Most studies (67%) used a validated measure of psychological factors.

Overall, self-efficacy was most consistently associated with adherence, depressive symptoms were associated with adherence more than half of the time, and fear of cancer recurrence and perceived personal control were seldom associated with adherence.

#### 3.3.4. Healthcare Provider-Related Factors

A total of 26 studies examined healthcare provider (HCP)-related factors for association with adherence (see Table 6) [25,29,31,32,41,44,45,49,50,51,56,60,62,65,67,70,71,72,73,75,79,81,84,85,86,87]. Quality of relationship with HCP was most consistently associated in univariate models (83% of the time) but less so (50% of the time) in multivariate models. Perceived supportiveness of HCPs, perceived self-efficacy in communicating with HCPs, and considering information received understandable were associated with adherence 100% of the time in univariate models but were examined by a small number of studies. Whether information (about AET) received was sufficient was examined several times but was seldom associated with adherence. Several additional factors were associated with adherence (e.g., trust in physician) but only examined in one study. No provider-related factors were examined often enough for Chi square testing. Almost half (46%) of studies used a validated measure of provider-related factors.

In sum, the quality of relationship with HCPs was most consistently associated with adherence and whether information received (about AET) was sufficient was not consistently associated.

#### 3.3.5. Sociocultural Factors

Fifteen studies [25,31,34,41,52,56,65,73,74,77,79,80,84,86,88] examined sociocultural factors (see Table 7). Social support was examined 13 times, associated with adherence most of the time (78%) in univariate models and over half the time in multivariate models. Studies that examined social support most often had prospective designs (*n* = 8) and objective measures of adherence (*n* = 8). Emotional support, material support, and perceived social norms regarding AET use were associated 75% or more of the time in univariate models in a small number of studies, but associations did not hold in multivariate models. One study examined whether AET adherence differed during Ramadan (which involves fasting), concluding there was no difference [88]. No sociocultural factors were examined frequently enough for Chi square testing. Fewer than half (40%) of studies used a validated measure to assess sociocultural factors.

In sum, social support was most consistently associated with adherence. A small number of studies supported an association between material support, emotional support, and perceived social norms with adherence.

#### 3.3.6. General and Quality of Life Factors

In total, 24 studies [27,29,34,41,43,44,46,47,50,55,56,57,58,59,64,67,71,73,80,81,82,89,90,91] examined general and quality of life-related factors, see Table 8. Quality of life/general well-being and physical functioning were examined most often (*n* = 7 each). However, no factor from this group that was examined more than twice was consistently associated with adherence. One retrospective study reported that a longer prescription interval was associated with both adherence and persistence in univariate and multivariate analyses. Several factors were associated (e.g., fertility concerns, pain before starting medication, having searched for medication online) or not associated (e.g., sexual functioning, forgetting, and health literacy) with AET adherence, but were only examined once. No general factors were examined often enough for Chi square testing. Over half (55%) of studies used a validated measure.

Overall, no general or quality of life factors that were assessed by two or more independent studies were consistently associated with adherence.

## 4. Discussion

Prior reviews indicate the potential importance of several factors in understanding adherence to AET, including but not limited to positive decisional balance, self-efficacy, necessity beliefs, social support, side effects, and relationships with HCPs [8,12,16,17]. The present review extends the literature by identifying which factors appear to be most consistently associated with adherence relative to other factors. Self-efficacy and positive decisional balance were the only factors examined ≥10 times and associated ≥75% of the time in both univariate and multivariate models (see Figure 2). Although these numerical cut-offs are arbitrary, this review suggests that self-efficacy and positive decisional balance in particular are consistently associated with adherence and thus merit further investigation as factors that could be targeted to support women in adhering to AET.

Results are consistent with prior studies suggesting a link between self-efficacy and medication adherence more generally. A systematic review of 154 studies reports that medication-specific self-efficacy, disease management self-efficacy, and general self-efficacy are associated with medication adherence across several chronic illness populations, including individuals with HIV/AIDS, cardiac and vascular disorders, and respiratory disorders [92]. Social cognitive theory posits that self-efficacy can be influenced by factors including prior accomplishment, vicarious learning, and verbal persuasion [93], which might be targeted by components such as collaborative problem solving with healthcare providers or other AET users, and education. A systematic review and meta-analysis of >100 studies indicates that experimentally induced changes in self-efficacy are associated with medium effect-size changes in health behaviors (e.g., exercise, physical activity, and condom use; d = 0.47) and intentions (d = 0.51) [94]. Systematic reviews also demonstrate that self-efficacy is modifiable in a healthcare context. For example, multimodal interventions (including elements such as goal setting, problem solving, education) have been used to increase self-efficacy for exercise among patients with heart failure [95] or self-efficacy for engaging in musculoskeletal rehabilitation [96].

Of note, nearly all studies in the present review that examine self-efficacy were cross-sectional and all used a self-report (vs. objective) measure of adherence. It is possible that associations between self-report adherence and self-efficacy may be in part due to shared reporting bias, where patients could be overestimating both. Nafradi et al.’s systematic review reports that associations between self-efficacy and medication adherence were more common in studies using subjective (87%) rather than objective measures (67%) of adherence [92]. They speculate that this may be in part explained by overlap between constructs, wherein some self-report adherence measures include questions about barriers that may relate to some aspects of self-efficacy [92]. Future studies that can demonstrate an association between self-efficacy and objectively measured AET adherence prospectively will provide further support for self-efficacy as an important factor in AET use.

Similar to prior studies [8,12,16,17], positive decisional balance was associated with AET adherence in studies of all designs and studies that used subjective and objective measures. Changing decisional balance involves either increasing perceived benefits and/or decreasing costs associated with one’s treatment decision. Knowledge about AET efficacy is necessary but may be insufficient for creating a positive decisional balance. For example, a possible reason prior interventions have been unable to improve AET adherence may be because they are largely based on education alone [10,11], and the present review reported that just receiving information about AET was rarely associated with adherence. Alternative methods of increasing perceived benefits or reducing perceived costs, such as motivational communication interventions, may be needed. Motivational communication involves a patient-centred interactional style that supports patients’ active role in change and acknowledges that both benefits and costs to change exist. It may also increase the perceived benefits of a behavior by linking it to one’s values and resolve ambivalence about the cost of change [97], which could in turn affect decisional balance. Motivational communication interventions have been demonstrated to improve adherence across several classes of medication. For example, a systematic review and meta-analysis of 17 RCTs of motivational interviewing for medication adherence (e.g., antiretrovirals, antihypertensives, antidepressants) reports the intervention to result in a 17% increased chance of being classified as adherent and an increase in continuously measured adherence (standardized mean difference = 0.70) compared to control groups [98].

Similar to prior reviews [8,14], side effects were not always associated with adherence. Only presence (vs. absence) of symptoms was examined more than 10 times and associated ≥50% of the time in both univariate and multivariate models. Several studies report that when women who discontinue medications are asked why, side effects are the most commonly reported reason [99,100,101,102]. Clearly, experiencing side effects does not invariably lead to poor adherence or discontinuation—rather the experience of side effects may be universal. For example, the prevalence of side effect presence was reported to be as high as 94% among the included studies, and the prevalence of arthralgia and vasomotor symptoms were reported to be as high as 88% and 96%, respectively. It may be the case that side effects interact with other variables (e.g., psychological distress) that when compounded, affect adherence. However, side effects may also be the easiest concrete thing for women to indicate when asked about discontinuation, and therefore are commonly reported. Some inconsistency as to whether side effects are associated with adherence may also reflect differences in when measures of adherence capture intentional or unintentional adherence, or when it may be ambiguous (e.g., when simply asking how many pills were missed in the past week). Future studies may help clarify this finding.

The present study reports that studies with prospective designs were more likely to report associations between side effects and adherence than cross-sectional or retrospective studies. The reason for this is unclear—one potential explanation may be early discontinuation in those experiencing the most bothersome side effects. For example, Lee et al. reported that adverse events (most often musculoskeletal pain) caused early discontinuation in a sample of 609 breast cancer survivors, half of the time [99]. Thus, those who experience the most severe side effects might not be captured in cross-sectional studies that recruit individuals who have been able to adhere for at least some time. Further research may determine whether this is a spurious result or a consistent pattern. Side effects were consistently reported to be associated with intentional non-adherence (characterized by deciding to miss a pill) and consistently not reported with unintentional non-adherence (typically characterized by forgetting), though this was represented by a small number of studies (*n* = 4).

A body of literature supports an association between depressive symptoms and lower adherence to medication for chronic illness [103,104]. However, depressive symptoms were not among the factors most consistently associated with adherence in the present study. Again, psychological distress following cancer, a life-threatening illness, is common [105], and may not invariably lead to non-adherence. Rather, depressive symptoms likely also interact with other factors in influencing adherence, such as social support and socioeconomic status. There are many evidence-based treatments for depressive symptoms in cancer survivors (e.g., Cognitive Behavioral Therapy, Mindfulness-Based Interventions), which could be recommended in these circumstances [106,107]. Future studies should incorporate multiple interacting factors into predictive models of adherence, which are more likely to represent the context in which breast cancer survivors take medications.

This review reported substantial variability in estimates of adherence (25.7–98%), which is consistent with the variability reported in previous reviews [8,13,108] and is likely due to heterogeneity in study designs, timeframes examined, measures used, and definitions of adherence. For example, as adherence decreases over time [8], higher rates of adherence could be found in cross-sectional studies recruiting women soon after AET initiation; adherence estimates could be lower if non-persistent women are also categorized as non-adherent; and HCP estimates of adherence could be inflated. The variability is particularly salient in the studies that reported disparate estimates of adherence across different measures used within the same sample (e.g., higher estimates based on self-report than based on prescription records) [26,27]. Heterogeneous estimates based on different measures underscores the importance of detailed reporting of how adherence is conceptualized, measured, and defined, as there are benefits and drawbacks to different measures of adherence. For example, self-report questionnaires allow convenience but are subjective to reporting biases while prescription records may provide more objectivity but may erroneously equate medications dispensed with medications consumed. Ultimately, multimethod approaches to adherence measurement (e.g., including self-report and objective measurement, taken at multiple time points and covering varying lengths of time) may better capture the nuances of how breast cancer survivors use AET and should be considered in future studies of AET adherence.

There are a number of limitations to the present study. First, all of the potentially modifiable factors in this study were examined in isolation, whereas in reality a complex interplay between factors is likely. For example, depressive symptoms, side effects, and social support could all interact to influence well-being. Furthermore, although sociodemographic factors such as income and insurance status were beyond the scope of the present review, they could have impacted the potentially modifiable factors examined. Future studies should examine interactional or other multivariable models to further our understanding of adherence. Second, although we made the effort to examine non-significant results, the likelihood of publication bias remains. Thus, the proportion of studies reporting associations between potentially modifiable factors and adherence could be inflated. Third, we were unable to thoroughly examine whether different factors were associated with adherence at various time points, though it is possible that factors might affect adherence differently across time. Finally, all of the studies in the present review were correlational and causation cannot be inferred. Therefore, there is no guarantee that the factors identified as most important by this review would successfully improve adherence if targeted by an intervention. For example, a large meta-analysis (*N* = 771) reported that interventions targeting behaviors are generally more effective for medication adherence than interventions targeting knowledge or beliefs [109].

## 5. Conclusions

Although AET reduces risk of breast cancer recurrence, adherence is suboptimal. The present systematic review examined potentially modifiable factors associated with AET adherence among breast cancer survivors. Self-efficacy and positive decisional balance were the factors most consistently associated with adherence. Although there is no guarantee that self-efficacy and positive decisional balance will be successful intervention targets (e.g., through components such as problem solving or motivational communication, respectively), this review suggests that they may be worthy of further investigation for this purpose, as prior interventions for AET adherence have been ineffective to date. Interventions that can consistently improve adherence to AET may have the potential to directly benefit the health and lifespan of breast cancer survivors.

## Figures and Tables

**Figure 1 cancers-13-00107-f001:**
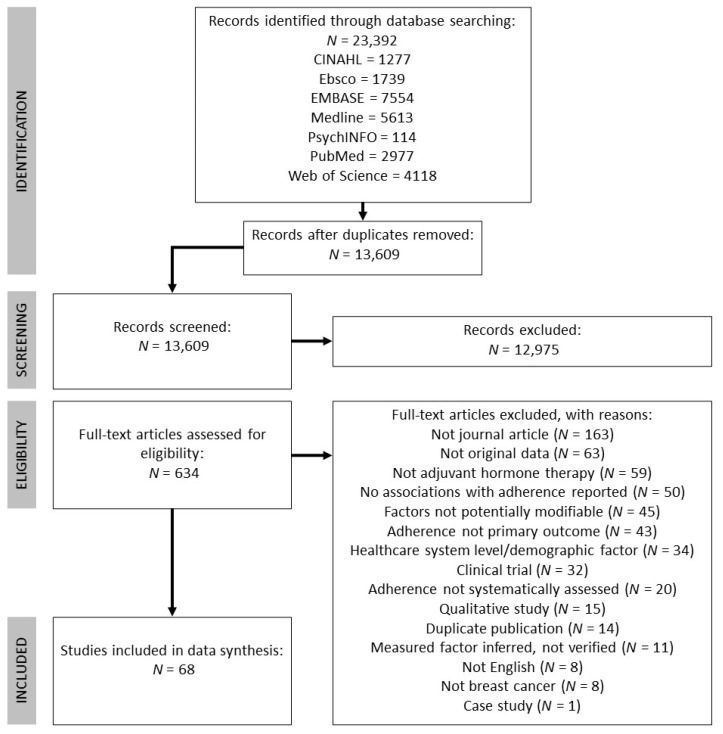
Flow chart detailing record identification, screening, and inclusion, adapted from PRISMA guidelines [19].

**Figure 2 cancers-13-00107-f002:**
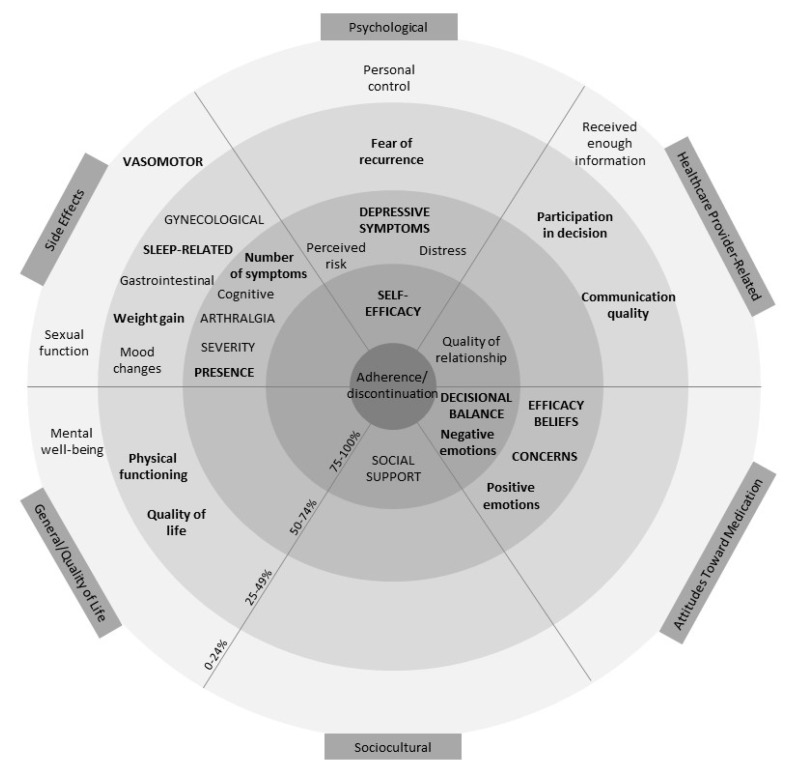
Visual representation of results. Within each of the six categories of factors identified, factors in closer proximity to the centre are associated with adherence a higher proportion of the time (0–24% in the outermost and 75–100% in the innermost ring). Factors in lowercase lettering have been examined ≥5 times, and factors in uppercase lettering have been examined ≥10 times. Bolding represents factors associated at least as often in multivariate models as in univariate models. Thus, factors in uppercase lettering, closest to the middle, and bold (i.e., self-efficacy and decisional balance) represent the most robust associations. Factors that were examined <5 times were not included in this graphic.

**Table 1 cancers-13-00107-t001:** Summary characteristics of all studies (*N* = 68).

Characteristic	M (Range)
Sample size (*N*)	815 (31–13,539)
Average age of sample (years)	57.41 (36.9–72.8)
Proportion who adhered, overall	74.82 (25.7–98.0)
Based on self-report	71.29 (25.7–95)
Based on objective measure	81.85 (42.0–98.0)
Based on physician report	81.13 (71.7–94.7)
Proportion who discontinued	23.46 (6.0–51.5)
Months on AETs ^a^	24.47 (4.5–36.0)
Number of associations examined	13 (1–91)
**Characteristic**	***N* (%)**
Sample Restrictions	
Non-metastatic/early-stage cancer only	34 (50.0%)
Older/post-menopausal women only	17 (25.0%)
Younger/pre-menopausal women only	4 (5.9%)
Recruitment Site	
Hospital/clinic	44 (64.7)
Clinical registry	11 (16.2)
Research registry	8 (11.8)
Insurance registry	4 (5.9)
Support group	1 (1.5)
AET Assessed	
Any AET	47 (69.1)
Tamoxifen only	11 (16.2)
Aromatase inhibitor only	10 (14.7)
Study Design	
Prospective	31 (45.6)
Cross-sectional	29 (42.6)
Retrospective	8 (11.8)
Outcome ^b^	
Adherence/non-adherence	55 (67.9)
Discontinuation/persistence	24 (29.6)
Composite of adherence and persistence	2 (2.5)
How Outcome was Measured ^b^	
Self-report	57 (70.4)
Objective	18 (22.2)
Physician report	5 (6.2)
Not reported	1 (1.2)
Variables selected based on theory/model of behavior change	16 (23.53%)

^a^ Not including studies where women are recruited at AET initiation. ^b^ Exceeds 68 as some studies examined multiple outcomes.

**Table 2 cancers-13-00107-t002:** Summary of quality indicators for all studies (*N* = 68).

Quality Indicator	Proportion Meeting Criteria	Whether Study Meets Criteria		
		*N* “Yes”	*N* “No”	*N* “Unclear”
Adherence or discontinuation clearly defined	89.71%	61	5	2
Complete outcome reporting	70.59%	48	4	16
Eligibility criteria clearly described	97.10%	66	2	0
Consecutive, random, or population-based sampling (vs. convenience sample)	44.12%	30	1	37
Statistical adjustment for potential confounds	69.12%	47	21	0
Control for multiple comparisons ^a^	69.70%	46	20	0

^a^ This criterion did not apply to two studies.

**Table 3 cancers-13-00107-t003:** Characteristics of studies that measured side effects (*N* = 44).

Side Effect Measure	Prevalence Range	*N* Studies Reported an Association/*N* Studies Examined	
Overall Measures		Univariate	Multivariate
Any measure of side effects ^a^	-	26/54	9/21
Side effect severity	-	12/22	2/12
Side effect presence (any)	26–94%	8/15	5/5
Number	-	3/5	1/2
Specific side effects			
Arthralgia	5–88%	9/17	2/5
Vasomotor/menopausal	3–96%	3/16	1/2
Gynecological	14–50%	4/10	0/2
Sleep-related	5–58%	4/10	1/2
Gastrointestinal	6–31%	3/7	0/1
Weight gain	31–54%	3/7	1/1
Cognitive	26–33%	5/7	0/1
Mood changes/anxiety	15–53%	3/7	-
Impact on sexual function	30–53%	0/5	-
Bladder control	24–29%	0/4	-
Vision issues	21–27%	1/3	-
Fluid retention/swelling	13–34%	1/2	-
Hair thinning/loss	14–35%	0/2	-
Headaches	16%	1/1	-
Pseudo-neurological	-	0/1	-
Shortness of breath	12–20%	0/1	-
Dizziness	18–19%	0/1	-
Breast sensitivity	22–24%	0/1	-
Bone-related changes	6–25%	0/1	-
Characteristics of Measure			*N* (%)
Self-report			41 (93)
Validated measure			24 (55)
Author created/not specified			16 (36)
Analysis of online discussions with healthcare provider			1 (2)
Physician reported			3 (7)

^a^ Composite variable created by authors for this review.

**Table 4 cancers-13-00107-t004:** Characteristics of studies that measured attitudes toward AET (*N* = 29).

Attitude-Related Factor	*N* Studies Reported an Association /*N* Studies Examined	
	Univariate	Multivariate
Belief in efficacy/necessity	12/20	8/9
Concerns over medication use	10/16	4/7
Positive decisional balance	8/10	6/9
Positive emotions/attitude	6/9	3/5
Negative emotions/attitude	5/5	4/4
Belief AET are harmful/overused	0/3	-
Intention to take AET	3/4	2/3
Attributing side effects to AET	3/4	0/2
Expected side effect severity	2/2	0/1
Characteristics of Measure		*N* (%)
Validated measure		12 (41)
Author-created or adapted measure		17 (59)

**Table 5 cancers-13-00107-t005:** Characteristics of studies that measured psychological factors (*N* = 30).

Psychological Factor	*N* Studies Reported an Association/*N* Studies Examined	
	Univariate	Multivariate
Depressive symptoms	9/14	5/7
Self-efficacy/perceived behavioral control	8/10	7/8
Fear of cancer recurrence	2/8	1/3
Personal control/internal locus of control	0/7	0/3
Perceived risk/susceptibility to recurrence	3/6	4/5
Anxiety/depression/distress	3/5	1/3
Anxiety symptoms	2/4	0/2
Perceived control over treatment	1/3	0/3
Coherence	1/3	1/3
Emotional representations	0/2	0/2
Self-efficacy for learning	1/2	1/1
Coping	0/2	0/1
Protection motivation	1/1	-
Optimism	1/1	-
Perceived cognitive function	1/1	1/1
Perceived aging	1/1	1/1
Perceived sensitivity to medicine	0/1	-
Characteristics of Measure		*N* (%) ^a^
Self-report		28 (93)
Author-created questionnaire		10 (33)
Validated measure used		20 (67)
Chart review (physician report)		1 (3)
Insurance claims database		1 (3)

^a^ Exceeds 100% as some studies used multiple measures.

**Table 6 cancers-13-00107-t006:** Characteristics of studies that measured healthcare provider-related factors (*N* = 26).

Healthcare Provider-Related Factor	*N* Studies Reported an Association/*N* Studies Examined	
	Univariate	Multivariate
Participation in decision to take AET	4/9	3/6
Information received was sufficient	2/9	0/3
Quality of relationship with HCP	5/6	2/4
Communication quality/frequency	2/5	1/1
Perceived supportiveness of HCPs	3/3	3/3
Perceived self-efficacy in communicating with HCPs	3/3	1/3
Information received understandable	2/2	1/2
Discussing side effects with HCPs	1/1	1/1
Value placed on physician’s opinion	1/1	1/1
Strength of physician’s recommendation	1/1	1/1
Perceived ageism from HCPs	1/1	-
Perceptions of physician’s knowledge of patient	1/1	-
Trust in physician	1/1	-
Perceived thoroughness of care	1/1	-
Consulting with HCPs when having trouble	0/1	-
Characteristics of Measure		*N* (%) ^a^
Author-created or not specified		15 (58)
Validated measure		12 (46)
Analysis of online portal discussions		1 (4)

^a^ Exceeds 100% as some studies used multiple measures.

**Table 7 cancers-13-00107-t007:** Characteristics of studies that measured sociocultural factors (*N* = 15).

Sociocultural Factor	*N* Studies Reported an Association /*N* Studies Examined	
	Univariate	Multivariate
Social support	10/13	4/7
Perceived social norms	4/4	1/3
Material support	3/4	0/1
Emotional support	3/3	1/2
Ramadan fasting	0/1	-
Characteristics of Measure		*N* (%) ^a^
Validated measure used		6 (40)
Author-created, modified, or not specified		9 (60)

^a^ Exceeds 100% as some studies used multiple measures.

**Table 8 cancers-13-00107-t008:** Characteristics of studies that measured general and quality of life factors (*N* = 24).

General/Quality of Life Factor	*N* Studies Reported an Association/*N* Studies Examined	
	Univariate	Multivariate
Quality of life/general well-being	2/7	2/3
Physical functioning	2/7	1/1
Mental well-being	1/5	-
Use of strategies to alleviate side effects	1/3	-
Use of strategies to help take medications	0/2	0/1
Longer prescription interval	2/2	2/2
Social functioning	1/2	1/1
Practical problems	1/2	1/1
Sexual interest/enjoyment	0/2	0/1
Sexual distress	0/1	-
Sexual functioning	0/1	-
Body image	0/1	-
Used books/magazines to help with decision	0/1	0/1
Used internet to help with decision	1/1	1/1
Searched for medication online (in general)	1/1	-
Forgetting	0/1	-
Health literacy	0/1	-
Fertility concerns	1/1	1/1
General barrier	1/1	1/1
Side effect-related barrier	1/1	-
Pain before starting medication	1/1	1/1
Characteristics of Measure		*N* (%) ^a^
Validated measure used		12 (55)
Author-created		6 (27)
Not specified/unclear		2 (9)
Online discussion portal		1 (5)
Pharmacy record		1 (5)

^a^ Exceeds 100% as some studies used multiple measures.

## Data Availability

No new data were created or analyzed in this study. Data sharing is not applicable to this article.

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
