# Peer review of "Potentially Modifiable Factors Associated with Adherence to Adjuvant Endocrine Therapy among Breast Cancer Survivors: A Systematic Review"

_cancers, 2020, doi:10.3390/cancers13010107_

Round 1

Reviewer 1 Report

The authors have done an excellent job summarizing the highly heterogeneous literature on potentially modifiable factors associated with adjuvant endocrine therapy (AET) use among breast cancer survivors. Their rigorous approach fills several gaps left by previous reviews on this topic, including an assessment of how the multiple ways in which adherence has been measured and type of study (e.g., prospective vs. cross-sectional) may relate to the type modifiable factors identified as statistically significant, as well as the consistency of those associations across multiple studies. The search strategy, data extraction, and analysis plan were well-described and should be reproducible. The results section is well organized and appropriately detailed, with Figure 2 serving as an especially helpful way to visualize the factors across multiple dimensions (frequency of report in the literature, frequency of association with adherence, and persistence of the association across univariate and multivariate models). The discussion is thorough, balanced, and thoughtfully nuanced.

There are just a couple of ways in which this manuscript could be improved:

1) Consider reformatting Table 1 to improve clarity and readability. The center justified left column (in this table and each of the following tables) is difficult to scan. In the lower half of this table, it would also be helpful to either bold or underline the characteristic group heading (e.g., Sample restrictions) and then indent each of the characteristics below the heading.

2) There are a few typos to correct – see lines 211-212 and 299.

Author Response

Thank you very much for your helpful comments on our manuscript. We are happy to make the requested revisions and agree that they will improve the quality of the manuscript. The changes are outlined subsequently.

Reviewer 1

1) Consider reformatting Table 1 to improve clarity and readability. The center justified left column (in this table and each of the following tables) is difficult to scan. In the lower half of this table, it would also be helpful to either bold or underline the characteristic group heading (e.g., Sample restrictions) and then indent each of the characteristics below the heading.”

Thank you for this suggestion. We have made these changes in all tables (table 1, page 6; table 2, page 7; table 3, page 9; table 4, page 10; table 5, page 11; table 6, page 12; table 7, page 12; & table 8, page 13)

2) There are a few typos to correct – see lines 211-212 and 299.”

Thank you for noting these. We edited lines 211-212 for clarity, they now read: “This aim examined whether a factor was more or less likely to be associated with adherence based on three methodological characteristics:…”. We similarly changed the wording of the subsequent sentence for clarity “It also examined whether a factor was more or less likely to be associated with adherence based on three sample characteristics: average age, adherence level, and time on AET” (line 216-218).

On line 301 (formerly 299) we changed “the” to “this”.

We reviewed the remainder of the document as well made the following corrections:

Line 44, changed to “focusing

Line 58, added “[8]

Line 229, added “years

Line 267, Table 2, “population

Reviewer 2 Report

The objective of this work is to provide a systematic, up-to date review of the potentially modifiable factors associated with Adjuvant Endocrine Therapy (AET) in breast cancer survivors.

The topic is not really original but the authors addressed and covered it very effectively, with the following main strengths:

  • A strong, robust methodology to perform the review, very well-described and in accordance with the international guidelines. Especially, the search strategy and the analysis plan are quite appropriate to properly select the relevant studies on one hand and to take into account the heterogeneity in study designs on the other hand. Therefore the potential factors identified in the review are highly supported by relevant parameters as the knowledge about the recording of proportion of results for univariate and multivariate analyses; and the description of the methodological characteristics of the studies designs and outcomes assessed.
  • A detailed and comprehensive presentation of the outlined factors, grouped into 6 categories on an attractive visual representation.
  • Some very interesting findings, especially about the low level of association between side effects or depressive symptoms, respectively, and adherence to treatment for chronic illness.
  • A relevant discussion to put into perspective the main results, especially the links between self-efficacy, positive decisional balance or side effects and treatment adherence, and to examine their consistence with prior studies. I fully agree with the issue that knowledge about AET efficacy could not be sufficient for creating a positive decisional balance to support treatment adherence. As well, the fact that studies with prospective designs are not likely to report associations between side effects and adherence than retrospective studies is clearly highlighted and discussed.

We noted certain weaknesses in the current draft. First, the introduction should be reduced in order to more directly present the background and the objectives of the review. Especially, the two paragraphs starting by “Several reviews…” and “Some existing reviews..” (lines 72 – 95) are too long and could be reduced. Secondly, some promising aspects could rather be developed or detailed, for instance the use of multi-modal interventions to increase self-efficacy or the use of multi-method approaches to adherence measurement. Finally, even if we could understand that as the authors said in their conclusion, “There is no guarantee that self-efficacy and positive decisional balance will be successful intervention targets”, we would like to better understand what these interventions might be in practice.

Author Response

Thank you very much for your helpful comments on our manuscript. We are happy to make the requested revisions and agree that they will improve the quality of the manuscript. The changes are outlined subsequently.

Reviewer 2

  1. We noted certain weaknesses in the current draft. First, the introduction should be reduced in order to more directly present the background and the objectives of the review. Especially, the two paragraphs starting by “Several reviews…” and “Some existing reviews..” (lines 72 – 95) are too long and could be reduced.”

Thank you for this suggestion. We have shortened these two paragraphs and combined them into one. It now reads “Several reviews identify that being unmarried, having more comorbidities, identifying as non-White, having later stage cancer, extremes of age, and higher cost for AET are associated with poorer AET adherence [12-16]; though these sociodemographic factors are non-modifiable. Multiple systematic and integrative literature reviews have been published that examine patient-reported and psychosocial factors associated with AET adherence or persistence in breast cancer survivors – they generally measure AET adherence and persistence via self-report or prescription records and define adherence as MPR ≥80% [8,12,13,16,17]. In reviews that only summarize significant relationships, factors associated with AET adherence include side effects, self-efficacy, belief in necessity of medications, social support, healthcare provider relationship, forgetfulness, and knowledge of cancer [12,16]. Reviews that also include null findings typically endorse social support, positive decisional balance, beliefs about medications, and self-efficacy as associated with adherence; but also indicate that patient-provider relationship or communication, depressive symptoms, and side effects are not always associated with adherence [8,17]. A 2015 meta-analysis indicates that side effect presence is associated with over five times the odds of discontinuing AET and nearly two times greater odds of non-adherence, however this meta-analysis includes only two studies [18].” (lines 72-96). 

We also shortened line 97 to read “There are notable limitations to existing systematic reviews”.

  1. Secondly, some promising aspects could rather be developed or detailed, for instance the use of multi-modal interventions to increase self-efficacy or the use of multi-method approaches to adherence measurement.”

To elaborate on multi-modal interventions to increase self-efficacy, we added the following: “Social cognitive theory posits that self-efficacy can be influenced by factors including prior accomplishment, vicarious learning, and verbal persuasion [93], which might be targeted by components such as collaborative problem-solving with healthcare providers or other AET users, and education” (line 424-427). In adding an additional reference we amended the subsequent references.

To elaborate on multi-method approaches to adherence assessment, we added the following: “(e.g., including self-report and objective measurement, taken at multiple time points and covering varying lengths of time)” (line 510-511).

  1. Finally, even if we could understand that as the authors said in their conclusion, “There is no guarantee that self-efficacy and positive decisional balance will be successful intervention targets”, we would like to better understand what these interventions might be in practice.”

Thank you for highlighting that this sentence needed more clarification. We agree and have modified this sentence to read: “Although there is no guarantee that self-efficacy and positive decisional balance will be successful intervention targets (e.g., through components such as problem-solving or motivational communication, respectively)…” (line 537-538). We hope that this change, which also refers to the previous sections of the discussion where we have elaborated on these specific intervention components (page 14, paragraph 2 & 4) provides clarification without adding redundant information.

Reviewer 3 Report

Authors have done a good job to review the existing literature on challenges with adjuvant endocrine treatment in breast cancer. It highlights the limitations in a conducting a proper meta analysis  of this issue.  They have acknowledged the limitations of this study well. 

Author Response

Thank you very much for your review and your comments.

Reviewer 4 Report

AET including the selective estrogen receptor modulator tamoxifen (TAM) and aromatase inhibitors (AIs), is well-established to reduce risk of hormone receptor-positive breast cancer recurrence. However, in practice adherence to AET is suboptimal. 

The present systematic review examined potentially modifiable factors associated with adherence to Adjuvant endocrine therapy (AET), in accordance with PRISMA guidelines (PROSPERO registration ID: CRD42019124200). 

This study extends the literature by identifying which factors appear to be most consistently associated with adherence relative to other factors. 
This is an interesting and useful study that contributes to better understanding the factors involved in adherence to AET and provides an unbiased overview of the body of knowledge on this specific topic.
The authors have been respectful with principles and methods behind of systematic reviews.

Recommended for publication. No amends are needed.

Author Response

(The authors gave the same response as above.)
